# Contribution of brown carbon on light absorption in emissions of European residential biomass combustion appliances

Satish Basnet[1], Anni Hartikainen[1], Aki Virkkula[2], Pasi Yli-Pirilä[1], Miika Kortelainen[1], Heikki Suhonen[1], Laura Kilpeläinen[1], Mika Ihalainen[1], Sampsa Väätäinen[1], Juho Louhisalmi[1], Markus Somero[1], Jarkko Tissari[1], Gert Jakobi[3,4], Ralf Zimmermann[3,4], Antti Kilpeläinen[5] and Olli Sippula[1,6]

[1]Department of Environmental and Biological Sciences, University of Eastern Finland, FI-70211, Kuopio, Finland
[2]Atmospheric Composition Research, Finnish Meteorological Institute, Helsinki, Finland
[3] Department of Analytical and Technical Chemistry, University of Rostock, DE-18059, Rostock, Germany
[4]Joint Mass Spectrometry Centre, Comprehensive Molecular Analytics, Helmholtz Zentrum München, DE-85764, Neuherberg, German
[5]School of Forest Sciences, University of Eastern Finland, FI-80101, Joensuu, Finland
[6]Department of Chemistry, University of Eastern Finland, FI-80101, Joensuu, Finland

*Correspondence to*: Satish Basnet (satish.basnet@uef.fi), Anni Hartikainen (anni.hartikainen@uef.fi), Olli Sippula (olli.sippula@uef.fi)

**Abstract.** Residential biomass combustion significantly contributes to light-absorbing carbonaceous aerosols in the atmosphere, impacting the Earth's radiative balance at regional and global levels. This study investigates the contribution of brown carbon to the total particulate light absorption in the wavelength range of 370 to 950 nm ($BrC_{370-950}$) and the particulate absorption Ångström exponents ($AAE_{470/950}$) in 15 different European residential combustion appliances using a variety of wood-based fuels. $BrC_{370-950}$ was estimated to be from 1 % to 21 % for wood log stoves and was primarily influenced by fuel moisture content and minimally affected by combustion appliance type. The $BrC_{370-950}$ contribution was 10 % for a fully automatized residential pellet boiler. Correlations between the ratio of organic to elemental carbon (OC/EC) and BrC indicated that a one unit increase in OC/EC corresponded to approximately a 14 % increase in $BrC_{370-950}$. Additionally, $BrC_{370-950}$ increased with decreasing combustion efficiency. $AAE_{470/950}$ of wood log combustion aerosols ranged from 1.06 to 1.61. By examining the correlation between $AAE_{470/950}$ and OC/EC, an $AAE_{470/950}$ close to unity was found for pure black carbon (BC) particles originating from residential wood combustion. This supports the common assumption used to differentiate light absorption caused by BC and BrC. Moreover, diesel aerosols exhibited an $AAE_{470/950}$ of 1.02, with $BrC_{370-950}$ contributing only 0.66 % to the total absorption, aligning with the assumption employed in source apportionment. These findings provide important data to assess the $BrC_{370-950}$ of RWC emissions with different emission characteristics and confirm that BrC can be a major contributor to particulate UV and near-UV light absorption for Northern European wood stove emissions with relatively high OC/EC ratios.

**Lists of abbreviations**

| | |
|---|---|
| $AAE_{470/950}$ | absorption Ångström exponent for the wavelength-pair 470 nm and 950 nm |
| $AAE_{370-950}$ | absorption Ångström exponent attained by fitting power law to the total absorption given by AE33 |
| $AAE_{BC}$ | absorption Ångström exponent for pure black carbon |
| AE33 | Aethalometer model 33 |
| $b_{abs}$ | absorption coefficient |
| BB | biomass burning |
| BC | black carbon |
| BrC | brown carbon |

| | |
|---|---|
| $BrC_{370-950}$ | contribution of BrC to the total absorption over wavelengths 370 nm to 950 nm |
| CMH | conventional masonry heater |
| DR | dilution ratio |
| eBC | equivalent black carbon |
| EC | elemental carbon |
| ED | Ejector diluter |
| EF | emission factor |
| GMD | geometric mean diameter |
| ILMARI | Aerosol physics, chemistry, and toxicology research unit |
| IMPROVE | Interagency Monitoring of Protected Visual Environments |
| MAC | mass absorption cross-section |
| $MAC_{aeth}$ | standard mass absorption cross-section applied by AE33 |
| $MAC_{880nm}$ | measured mass absorption cross-section of BC (880 nm) |
| $MAC_{OC}$ | measured mass absorption cross-section of the organic fraction |
| MCE | modified combustion efficiency |
| MCS | modern chimney stove |
| MMH | modern masonry heater |
| NIOSH | National Institute for Occupational Safety and Health |
| NrDE | non-road diesel engine |
| OA | organic aerosol |
| OC | organic carbon |
| PB | pellet boiler |
| PM | particulate matter |
| RWC | Residential wood combustion |
| SIMO | residential wood combustion simulator |
| SS | sauna stoves |
| STD | standard deviation |
| UV | ultraviolet |

## 1 Introduction

Atmospheric aerosols play a crucial role in various atmospheric processes and have direct and indirect effects on the Earth's radiative forcing and energy budget. Among these aerosols, black carbon (BC) is a significant component that contributes to global warming due to its light-absorbing properties (Bond et al., 2013; IPCC, 2014). Here, we determine BC as the optically defined fraction of soot (Petzold et al., 2013). It is primarily emitted into the atmosphere through incomplete combustion of fossil fuels, biofuels, and biomass burning (BB), including activities such as forest fires, residential combustion, engines, and industrial processes (Klimont et al., 2017). In addition to BC, certain fractions of organic aerosols, known as "brown carbon" (BrC), exhibit significant light absorption in the visible and near-ultraviolet (UV) wavelengths (Kirchstetter et al., 2004), thereby playing a role in radiative forcing (Cappa et al., 2019; Liu et al., 2015). According to numerous studies, low-temperature biomass and biofuel burning, as well as small-scale residential combustion are the primary sources of atmospheric BrC (Cappa et al., 2019; Pokhrel et al., 2017; Saleh et al., 2014; Cheng et al., 2011; Li et al., 2019b).

Residential wood combustion (RWC) is a significant contributor to the energy supply in Europe and many other parts of the world. Unfortunately, RWC's sustainability is hindered by its high emissions of particulate matter (PM) and gases, which have detrimental effects on climate, human health, and the environment. In the European Union (EU), small-

scale combustion is recognized as a primary source of particulate pollution, accounting for over 45% of total fine particulate matter ($PM_{2.5}$) emissions (EEA, 2013). In Finland, RWC is the largest source of BC emission and is estimated to contribute up to 55 % to the total BC emitted (Savolahti et al., 2016). However, accurately quantifying the direct radiative forcing of absorbing aerosols emitted from RWC is challenging due to gaps in knowledge regarding the particle optical properties and BC and BrC emission factors (EFs).

Global simulations have indicated that BrC contributes approximately 10–50 % of the total absorption by atmospheric aerosols (Feng et al., 2013; Bahadur et al., 2012; Lack et al., 2012). In the case of biomass and wood log combustion, BrC has been found to contribute up to 20–90 % of light absorption in fresh RWC emissions at shorter visible wavelengths (Fang et al., 2022; Martinsson et al., 2015). However, only a few RWC appliance types and fuels have been studied (e.g., Fang et al., 2022; Martinsson et al., 2015; Zhang et al., 2020; Saleh et al., 2013; Kumar et al., 2018; Saliba et al., 2018; Olsen et al., 2020; Li et al., 2022; Leskinen et al., 2023; Tissari et al., 2019; Hartikainen et al., 2020; Savolahti et al., 2016, 2019; Sun et al., 2021), which causes uncertainties in assessing the direct radiative forcing of RWC emissions. In addition, the optical properties of emissions undergo changes in the atmosphere, which adds further complexity to estimating their climate effects (Laskin et al., 2015; Kumar et al., 2018; Zhong and Jang, 2014; Wang et al., 2019; Lack et al., 2012).

For the quantification of BrC, various optical measurement techniques have been employed (Fang et al., 2022; Pani et al., 2021; Pokhrel et al., 2017; Olson et al., 2015; Martinsson et al., 2015; Bahadur et al., 2012; Zhang et al., 2020; Li et al., 2022; Sun et al., 2021; Zhang et al., 2021; Rathod and Sahu, 2022; Massabò et al., 2015). The methods commonly involve the determination of the wavelength-dependent light absorption and subtracting the inferred BC absorption at specific wavelengths. The wavelength dependence of aerosol light absorption is generally described by the absorption Ångström exponent (AAE, Eq. 1) (Fang et al., 2022; Olson et al., 2015; Bahadur et al., 2012; Martinsson et al., 2015) based on a selected wavelength pair.

$$AAE_{\lambda_1/\lambda_2} = -\frac{\ln(b_{abs,\lambda_1}/b_{abs,\lambda_2})}{\ln(\lambda_1/\lambda_2)} \qquad (1)$$

BC exhibits strong light absorption across the visible spectrum, which typically results in an AAE of approximately 1.0. Higher AAE values are often assumed to indicate the presence of BrC. However, AAEs in the range of 0.8–1.4 have been measured and modelled even for pure BC particles (Liu et al., 2018; Virkkula, 2021), introducing uncertainty when determining BrC absorption (Lack and Langridge, 2013). Furthermore, the use of aethalometers, common instruments for real-time measurements of BC particles in ambient air, introduces additional uncertainties to BC and BrC quantification. The aethalometer measures the optical attenuation by aerosol particles collected on a filter, and converting the measured infrared light attenuation to the mass concentration of equivalent black carbon (eBC) requires knowledge of the mass absorption cross-section (MAC) of the particles and the multiple scattering factor (C) in the filter (Weingartner et al., 2003; Drinovec et al., 2015; Sandradewi et al., 2008; Virkkula et al., 2007). Typically, the MAC and C predefined by the instrument manufacturer are applied. However, various experiments using alternative instruments in conjunction with an aethalometer have shown higher C values for biomass combustion and ambient air aerosols than those proposed by the instrument manufacturer (Kumar et al., 2018; Tasoglou et al., 2017; Yus-Díez et al., 2021; Backman et al., 2017).

In this study, we define BC and BrC emissions from a range of different Northern European wood combustion appliances using wavelength-dependent aerosol absorption data from a 7-wavelength aethalometer and thermal-optical

carbon analysis of particle samples. The contribution of BC and BrC to the total light absorption is determined considering both the whole aethalometer wavelength range and the specific wavelengths. We investigate the effects of combustion technologies and fuel properties on the particulate AAE and BrC light absorption. Finally, we correlate particulate composition and combustion conditions with AAE and BrC absorption and establish parametrizations to evaluate the contribution of BrC to particle light absorption in RWC emissions.

## 2 Methods

The combustion experiments were performed in two facilities at the University of Eastern Finland using 7 different fuels and 15 different combustion appliances in total. Firstly, masonry heater, chimney stove, pellet boiler, and non-road diesel engine experiments were performed at the ILMARI laboratory (www.uef.fi/ilmari). Secondly, measurements with the masonry heater and 10 types of sauna stoves were conducted at the small-scale combustion simulator (SIMO) (www.uef.fi/en/web/fine/simo; Tissari et al., (2019)). The emission sources along with the fuels and details of measurement campaigns are summarized in Table 1. Details of the experimental setups, fuels in use, and basic emission characterization have been previously published elsewhere (see references in Table 1) for all experiments except for a subsection of experiments performed with the pellet boiler and the non-road diesel engine (ILMARI 2016), and the modern chimney stove (ILMARI, 2021). The following sections briefly explain the experimental setups for emission sources. The fuel compositions are shown in Supp. Table S1 and the experimental setups at the ILMARI and SIMO facilities are illustrated in Supp. Figure S1.

**Table 1.** Appliance types, fuel burnt, fuel moisture content, and fuel batch details for measurement campaigns under investigation.

| Appliance | Abbrv. | Measurement site and date | fuel type | fuel moisture (%) | number of repetitions | number of batches | time per batch (min) | average batch size (kg) | Reference |
|---|---|---|---|---|---|---|---|---|---|
| Modern masonry heater | MMH | ILMARI 2013 | beech | 9 % | 4 | 6 | 30 | 2.5 | Czech et al., 2018; Kortelainen et al., 2018 |
| | | | spruce | 7.4 % | 3 | 6 | 30 | 2.5 | |
| | | | birch | 7.2 % | 6 | 6 | 30 | 2.5 | |
| Conventional masonry heater | CMH | SIMO 2019 | birch | 11 % | 3 | 3–4 | 25 | 1.5 | Suhonen et al., 2021 |
| Sauna stoves (10 types) | SS-11 | SIMO 2018 | birch | 11 % | 23 | 3 | 15–45 | 3 + 3 + 1 | Tissari et al., 2019 |
| | SS-18 | | birch | 18 % | 12 | 3 | 15–45 | 3 + 3 + 1 | |
| | SS-28 | | birch | 28 % | 3 | 3 | 15–45 | 3 + 3 + 1 | |
| Modern chimney stove | MCS | ILMARI 2016 | pine | 6.1 % | 3 | 5 | 35–45 | 2 | Ihantola et al., 2020 |
| | | | spruce | 7.4 % | 3 | 5 | 35–45 | 2 | |
| | | ILMARI 2021 | beech | 9 % | 14 | 6 | 35 | 2 | |
| | | | peat | 17.9 % | 3 | 8 | 25 | 1 | |
| Pellet boiler | PB | ILMARI 2016 | softwood pellets | 7.3 % | 3 | 4 | 60 | | |
| Non-road diesel engine | NrDE | ILMARI 2016 | diesel | | 3 | 4 | 60 | | |

### 2.1 Combustion Appliances and Fuels

#### 2.1.1 Modern masonry heater

The modern masonry heater (MMH) (model HIISI, Tulikivi Ltd, Finland) was used with three different wood species (beech [*Fagus sylvatica*], birch [*Betula pubescens*], and spruce [*Picea abies*]) at the ILMARI facility of the University of Eastern Finland. Each experiment lasted for four hours with six batches of wood combustion, each burning for approximately 30 minutes. After the final batch was consumed the embers were stoked, and secondary air channels were closed for residual char combustion measurement. The total amount of wood combusted was approximately 15 kg per experiment (~ 2.5 kg per batch). The detailed experimental setups and procedures, along with the fuel properties, and emission characterization can be found in Czech et al., (2018) and Kortelainen et al., (2018).

#### 2.1.2 Conventional masonry heater

The Finnish logwood-fired conventional masonry heater (CMH) designed for laboratory studies was used in the SIMO with birch as fuel. Each combustion experiment had three batches and a total of approximately 4.65 kg (1.65 kg + 1.5 kg + 1.5 kg) of fuel was burnt. The detailed experimental setup, fuel properties, and emission characterization can be found in Suhonen et al., (2021).

#### 2.1.3 Sauna stove

Wood burning in sauna stoves (SS) was extensively measured in the SIMO facility. The typical use of sauna stoves in Finland was simulated using 10 different types of commercially available SS. Three batches of birch were combusted with a total load of approximately 7 kg (3 kg + 3 kg + 1 kg). The differences regarding the 10 SS types, fuel properties, and emission characterization are discussed in detail by Tissari et al. (2019). The major differences regarding the stoves were that SS1–4, 8, and 9 were traditional steel stoves, SS5 had long flue gas ducts at both sides of the stove, and SS6, 7, and 10 had stones covering the stove's outer shell. SS1 and SS8 specifically used fuels with different moisture content, allowing the study of the impact of fuel moisture content on the emissions. The fuel moisture contents in use for the experiments were 11 %, 17 % and 18 %, and 28 %. For this work, we have classified the SS results according to the fuel moisture contents, i.e., the results were averaged accordingly as SS-11 with 11%, SS-18 with 17 % and 18 %, and SS-28 with 28 % fuel moisture content because of the similarity of the optical properties between the SS types when using similar fuel moisture contents.

#### 2.1.4 Modern chimney stove

The modern chimney stove (MCS, Aduro, model 9.3, Denmark) was used in two different experimental campaigns in the ILMARI facility. In 2016, this non-heat retaining MCS was used to burn two different wood species (pine [*Pinus sylvestris*] and spruce). Every combustion experiment included five batches each containing approximately 2 kg of wood logs. After the combustion of the final batch, the embers were stoked, and emissions were measured from the residual char with the secondary air channel closed. An experiment lasted for 4 hours with each batch burning for approximately 35–45 minutes. Detailed information about the experimental setups, fuel properties, and the characterization of emissions can be found in Ihantola et al. (2020).

The second set of experiments using MCS was conducted in 2021 in the ILMARI facility using beech logs as fuel. Each combustion experiment had six batches with individual batches consisting of approximately 2 kg of wood logs.

The embers were stocked after the final batch was consumed, and the secondary air channel was closed during the residual char combustion.

The MCS was further used to burn peat briquets in the set of experiments conducted in 2021 in the ILMARI facility. The peat combustion experiment lasted for 4 hours, with eight 1 kg batches each burning for approximately 25 minutes. The first batch of fuel was added to the smoldering embers of beech logs to ease the ignition of peat, which led to flaming conditions for the whole peat-burning experiment.

#### 2.1.5 Pellet boiler

A modern small-scale (25 kW) pellet boiler (PB) (model PZ- RL, Biotech Energietechnik GmbH, Austria) was used with softwood pellets as fuel in the ILMARI facility. The stove's detailed description can be found in Lamberg et al., (2011a). The PB was operated under four different load capacities: low-load (7 kW), half-load (12.5 kW), high-load (18.5 kW), and full-load capacity (25 kW). The combustion lasted for 4 hours, with each mode operating for an hour before switching to the next load in ascending order.

#### 2.1.6 Non-road diesel engine

An EPA Tier 1/EU Stage II water-cooled non-road diesel engine (NrDE) (Kubota D1105-T, 24.5 kW, 3000 rpm, total displacement 1123 $cm^3$, Kubota, Europe) was used in the ILMARI facility in 2016. The engine was operated using a four-stage test cycle, for which the operating modes were selected based on the C1 test type of the ISO 8178 standard. The operating modes were idle (880 rpm), intermediate 50 % (2 050 rpm, 39.92 Nm, 8.57 kW), rated 10 % (3 000 rpm, 7.07 Nm, 2.22 kW), and rated 50 % (3 000 rpm, 35.33 Nm, 11.10 kW). The load of the engine was controlled with a liquid-cooled eddy current engine dynamometer (Froude Consine Inc., AG30HS, USA), capable of producing a maximum power of 30 kW and a maximum torque of 90 Nm (14 000 rpm). The test fuel was European commercial diesel (EN 590). Each experiment lasted for 4 hours, with each cycle operating for an hour before switching to the next cycle in the above-mentioned order.

### 2.2 Aerosol measurements

Aerosol light absorption and equivalent black carbon (eBC) mass concentration were measured using a dual spot aethalometer (AE33, Magee Scientific). The instrument uses two sample spots with different flow rates for aerosol accumulation and an unloaded spot as a reference area for calculating the light absorption coefficients ($b_{abs}(\lambda)$). The instrument measures light attenuation at seven wavelengths (370 nm, 470 nm, 520 nm, 590 nm, 660 nm, 880, and 950 nm). The AE33 automatically compensates for the filter-loading parameter (k) and multiple scattering coefficient (C) in real-time (Drinovec et al., 2015, 2017). In experiments prior to 2021, a filter tape made of TFE-coated glass fibers (M8020) was used for AE33. For this filter tape C value of 1.57 and a leakage factor of 0.07 were applied as provided by the manufacturer (Drinovec et al., 2015). For the ILMARI 2021 experiments, a newer filter tape (M8060, made of glass fibers coated with PTFE/PET) with a recommended/assumed C value of 1.39 and a leakage factor of 0.01 was used.

Two aethalometer instruments were in use: 'AE33a' in the ILMARI facility, and 'AE33b' in the SIMO facility. During the MCS experiments in ILMARI (2021), the two AE33s were utilized in line to compare the instruments' efficiency and reliability. Low variation (AE33a/AE33b 0.87 ± 0.09 on average ± standard deviations of the average values of the four experiments for the whole experiments, and 0.89 ± 0.09 for batch-wise averages excluding the ember phase) was observed in the measured eBC values by both AE33s, validating the reliability of AE33 data. Time series for

eBC emission concentration derived simultaneously from the two instruments (AE33a and AE33b) for two experimental days are presented in Supp. Figure S2.

The particle size distributions were measured with an Electric Low-Pressure Impactor (Classic ELPI, Dekati Ltd.). Organic (OC) and elemental (EC) carbon were collected on quartz fiber filters (Pallflex Tissuquartz) and analyzed with a thermal-optical method (Turpin et al., 2000) using a Carbon analyzer (Sunset Laboratory Inc.). The National Institute for Occupational Health (NIOSH 5040) protocol (NIOSH, 1999) was used for all experiments except for ILMARI (2021), where the analyses were based on the Interagency Monitoring of Protected Visual Environments (IMPROVE-A) protocol (Chow et al., 2007). Both protocols have been found to reliably provide the OC and EC concentrations (Wu et al., 2016). However, the IMPROVE-A protocol provides slightly more detailed information about OC and EC fractions at different temperatures than the NIOSH5040 protocol. Further, IMPROVE-A includes a reflectance-based optical correction that enables the analysis of heavily loaded filters.

### 2.3 Sampling

The raw flue gas sample was directed to the gas analyzers through an insulated and heated (180 °C) sample line with a ceramic filter. In the ILMARI setup, gaseous emissions ($CO_2$, CO, and $NO_x$,) were measured by an ABB Hartman & Braun (ABB Cemas Gas Analysing Rack, ABB Automation GmbH) and Siemens (ULTRAMAT 23) gas analyzer systems (Supp Figure S1). Similarly, the SIMO setup used Siemens ULTRAMAT 23 gas analyzers to measure $CO_2$, CO, and NO in the undiluted exhaust.

For the particle measurements, the sample was led through a two-stage dilution system. The first stage of dilution was performed with a porous tube diluter where the sample was diluted with clean air of ambient temperature followed by an ejector diluter (ED) (DAS, Venacontra). In the ILMARI, the dilution ratio (DR) after the first two stages of the dilution was 10–100 (Supp. Table S2). Additional ED (Palas, VKL 10 E) was used upstream of the aethalometer to further dilute the sample by an additional factor of 100 to reach suitable concentrations for aethalometer measurements. In SIMO, the initial DR after two stages of dilution was ~ 73–90 and an additional ED (DI-1000, Dekati Ltd.) was used upstream of the aethalometer to dilute the sample by an additional factor of 8.6. DR was defined from $CO_2$ concentrations in the raw gas and in the diluted sample (ABB $CO_2$ analyzer or Vaisala GMP 343 in ILMARI or Vaisala GMP 343 in SIMO) as described by Sippula et al. (2009).

### 2.4 Data analysis

#### 2.4.1 Segregation of black and brown carbon

The contribution of black and brown carbon to the total absorption of visible light were segregated by calculating 'dimensionless integrated absorptions' (DIA) of BC and BrC over the measured wavelength range. This approach relies on the multi-wavelength attenuation measurement and is comparable to the multi-wavelength absorption analyzer model (Massabò et al., 2015; Bernardoni et al., 2017). First, the dimensionless integrated absorption by BC ($DIA_{BC}$) was calculated by a definite integral of Eq. (2) for the aethalometer wavelength range (370–950 nm) using the absorption coefficient value at the wavelength of 880 nm ($b_{abs}$(880 nm)), where the light absorption was assumed to be the result of BC. The BC absorption was extrapolated to lower wavelengths assuming that BC has an AAE ($AAE_{BC}$) of 1.

$$b_{abs,BC}(\lambda) = b_{abs}(880nm) \times \left(\frac{\lambda}{880}\right)^{-AAE_{BC}} \tag{2}$$

For the calculation of the total absorption, a power law function was fit to the total aerosol absorption coefficients measured at the seven aethalometer wavelengths using the method of least squares. As a result, we attain an AAE describing the wavelength dependency over the visible wavelength range ($AAE_{370\text{-}950}$). The total dimensionless integrated absorption ($DIA_{Tot}$) was then calculated as the definite integral of this power law function over the aethalometer wavelength range. As the total absorption can be considered to be the sum of the absorptions by BC and BrC, the dimensionless integrated absorption by BrC ($DIA_{BrC}$) was defined as the difference between the $DIA_{Tot}$ and the $DIA_{BC}$. The relative contribution of BrC to the total absorption over wavelength 370 nm to 950 nm ($BrC_{370\text{-}950}$) was calculated as the ratio of $DIA_{BrC}$ to $DIA_{Tot}$. A step-by-step example of this calculation is shown in Supp. Section S1.

The absorptions by BC and BrC were also segregated for the individual wavelengths, for which the absorption by BrC was estimated as the difference between the total absorption and the absorption by BC (Olson et al., 2015). Major uncertainties in the relative contribution by BrC may arise due to the choice of value for $AAE_{BC}$, which may vary with differences in combustion emission. As the exact AAE for pure BC cannot be obtained with this instrumentation, we used the standard $AAE_{BC}$ of 1.0. Despite the uncertainties associated with this approach (Lack and Langridge, 2013), many studies have found the results in agreement with other methods of apportioning BC and BrC absorption (Tian et al., 2020; Zhang et al., 2019; Briggs and Long, 2016). The sensitivity of $BrC_{370\text{-}950}$ to the choice of $AAE_{BC}$ depends on the amount of BrC, as illustrated in Supp. Figure S12 and S13 for $AAE_{BC}$ values of 0.9 or 1.1 instead of 1.0. The results are, however, presented for an $AAE_{BC}$ of 1.0 with a brief discussion on the effects of $AAE_{BC}$ on the resolved absorption by BrC available in Section 3.2.

The eBC mass concentrations were converted from the $b_{abs}$(880nm) by multiplying it with the generally assumed MAC of eBC at 880 nm ($MAC_{aeth}$, 7.77 $m^2\ g^{-1}$) (Eq. 3).

$$eBC = b_{abs}(880\ nm)\ /MAC_{aeth}(880nm) \quad (3)$$

The actual MACs may diverge from the default $MAC_{aeth}$ with different emission sources (e.g., Olson et al., 2015). To assess the MAC of the emitted BC particles, the $MAC_{880nm}$ was estimated by dividing the measured $b_{abs}$ by the thermo-optically derived EC concentration (Eq. 4).

$$MAC_{880nm} = \frac{b_{abs}(880nm)}{EC} \quad (4)$$

Furthermore, the MAC of the organic fraction ($MAC_{OC}$) was estimated from the ratio between $b_{abs,\ BrC}$(470 nm), and the concentration of OC from the thermal-based method (Eq. 5).

$$MAC_{OC,470nm} = \frac{b_{abs,BrC}(470nm)}{OC} \quad (5)$$

The two-wavelength $AAE_{470/950}$ values typically used for source apportionment of ambient aerosol were calculated using the wavelength pair 470 nm and 950 nm (Eq. 6) as suggested by (Zotter et al., 2017).

$$AAE_{470/950} = \frac{-\ln(b_{abs,470\ nm}/b_{abs,950nm})}{\ln(470nm/950nm)} \quad (6)$$

**2.4.2 Emission factors**

The emission factors (EFs) were calculated as mg of PM (OC, EC, or eBC) per megajoule of energy produced applying Eq. 7 (Reda et al., 2015):

$$EF = c_n \times \lambda \times k \times Q_s \quad (7)$$

Where $c_n$ is the dilution corrected concentration of species in flue gas (mg $m^{-3}$), λ is the air-to-fuel ratio during the combustion process calculated from either raw flue gas $O_2$ concentration (ILMARI experiments, Eq. 8) or raw flue gas $CO_2$ concentration (SIMO experiments, Eq. 9). k is the correction factor for the fuel moisture content given by Eq. (10), where $H_u$ is the net heating value of dry fuel and $H_w$ is the amount of heat consumed in water evaporation which depends on the ratio of water to dry substance in fuel ($W_v$) and evaporation heat of water ($l_v$) (2.50 MJ $kg^{-1}$ at 0 °C, Eq. 11). The net heating values for the dry fuels used in different experiments are listed in Supp. Table S1. $Q_s$ is the dry volume of flue gas produced in the combustion of dry fuel (0.25 $m^3$ $MJ^{-1}$ for solid fuels and 0.26 $m^3$ $MJ^{-1}$ for diesel fuel).

$$\lambda = \frac{20.9\,\%}{20.9\,\% - O_2\%} \tag{8}$$

$$\lambda = \frac{20.2\,\%}{CO_2\%} \tag{9}$$

$$k = \frac{H_u}{H_u - H_w} \tag{10}$$

$$H_w = W_v \times l_v \tag{11}$$

The modified combustion efficiency (MCE, Eq. 12) was calculated to define the degree of completeness of a combustion process.

$$MCE = \frac{\Delta CO_2}{(\Delta CO_2 + \Delta CO)}, \tag{12}$$

where ΔCO and $\Delta CO_2$ are the background-corrected CO and $CO_2$ values in the exhaust.

### 2.4.3 Data handling and uncertainty estimation

The mean EFs for each fuel-appliance combination were calculated from the experiment-wise means averaged over the whole measurement period. These experiment-wise average EFs were further averaged according to the number of repetitions for each fuel type in individual combustion appliances (n = 3–23, Table 1), and results are given as mean EFs ± standard deviations (STD) of the repetition means. The $AAE_{470/950}$ and BrC contributions were calculated from the combustion period averaged $b_{abs}(\lambda)$ coefficient and further averaged according to the number of repetitions. The STD of $AAE_{470/950}$ is given as the mean of standard deviations of the time-resolved data. The STD of $MAC_{880nm}$ was calculated from $MAC_{880nm}$ values for each fuel type in individual combustion appliances.

The uncertainties related to the $AAE_{470/950}$ and the BrC absorption contribution depend on rates of change and uncertainties related to the attenuation and loading compensation parameters. The uncertainty in $AAE_{470/950}$ was derived as in Helin et al. (2021), and an overview of the estimated average relative uncertainty in $AAE_{470/950}$ over 1 minute, individual batches, and the complete combustion experiments are available in Supp. Table S3.

A detailed explanation regarding the uncertainty of the BrC absorption contribution is given in Supp. Section S2. The change in attenuation during each experiment was considered for the complete experiment, except for peat combustion, for which individual batches were considered instead. The absolute uncertainties in the fractions of absorption by BC and BrC are shown in Supp. Figure S4 for individual wavelengths for each fuel-appliance combination. Among RWC, sauna stoves (SS-11 and SS-18) showed the highest amount of absolute uncertainty in BrC absorption with a median value of ~ 7 % and ~ 10 % respectively for all calculated wavelengths, while MCS showed the least absolute uncertainty (~ 0.6 %). For the overall experiments, peat combustion showed the least amount of uncertainty with a median value of 0.01 % BrC contribution (470 nm). However, since the values for peat combustion were calculated from the

individual batches, the uncertainty values are not fully comparable to those of the other combustion sources. When considering the experiment-to-experiment variation in the uncertainty of the results, the diesel engine showed the least absolute uncertainty (median uncertainty ~ 0.3 % BrC contribution). However, the relative uncertainties in the contribution of BrC to the absorption were the highest when the BrC contribution was low, namely, for diesel exhaust emissions (0.66 %).

## 3 Results and discussion

### 3.1 Emission factors and combustion conditions

The EFs for OC, EC, and eBC (average ± standard deviation of the experiment-wise means) are presented in Figure 1. Among wood log combustion appliances, the highest eBC EF was observed for pine combustion in MCS (80.9 ± 45.7 mg $MJ^{-1}$), while the lowest was recorded for spruce combustion in MMH (27.9 ± 4.41 mg $MJ^{-1}$). Birch combustion in SS-11 (61.3 ± 34.4 mg $MJ^{-1}$) and birch combustion in CMH (59.4 ± 14.5 mg $MJ^{-1}$) exhibited the highest EFs for EC, whereas the lowest EC EF was found for spruce combustion in MMH (16.9 ± 1.23 mg $MJ^{-1}$). Good correlations were observed between EC and eBC concentrations within the fuel/appliance combinations (Supp. Figure S5). However, there was a disparity between the relations of eBC and EC between the different combustion fuel/appliance combinations, suggesting variations in the optical properties of emitted particles, as discussed further in Section 3.2.

The lowest OC EFs were observed for MMH, regardless of the wood species used. The formation of OC was influenced by fuel moisture, with the highest OC EF observed for high moisture content fuel combustion in SS-28 (51.8 ± 10.0 mg $MJ^{-1}$). Increased fuel moisture content decreases the temperature during RWC, which leads to increased emissions of particulate organic matter (Price-Allison et al., 2021, 2019; Magnone et al., 2016). The OC/EC ratio in the exhaust from sauna stoves increased from 0.51 ± 0.42 for 11 % moisture content to 1.49 ± 0.16 for 28 % moisture content. A moderate correlation was observed ($R^2 = 0.67$) between the OC/EC ratio and fuel moisture content for wood log fuel-appliance-wise averaged combination (Supp. Figure S6a). The presented EFs for EC, eBC, and OC from logwood appliances primarily represent typical Northern European wood stoves and generally align with similar studies conducted in Europe (Savolahti et al., 2016; Vicente and Alves, 2018) and North America (Tasoglou et al., 2017). In comparison to wood log combustion, pellet burning exhibited significantly lower EFs for OC, EC, and eBC (1.01 ± 0.25, 7.08 ± 3.29, and 8.03 ± 1.62 mg $MJ^{-1}$, respectively), supporting previous findings that pellet burners are relatively clean appliances (Lamberg et al., 2011a, b).

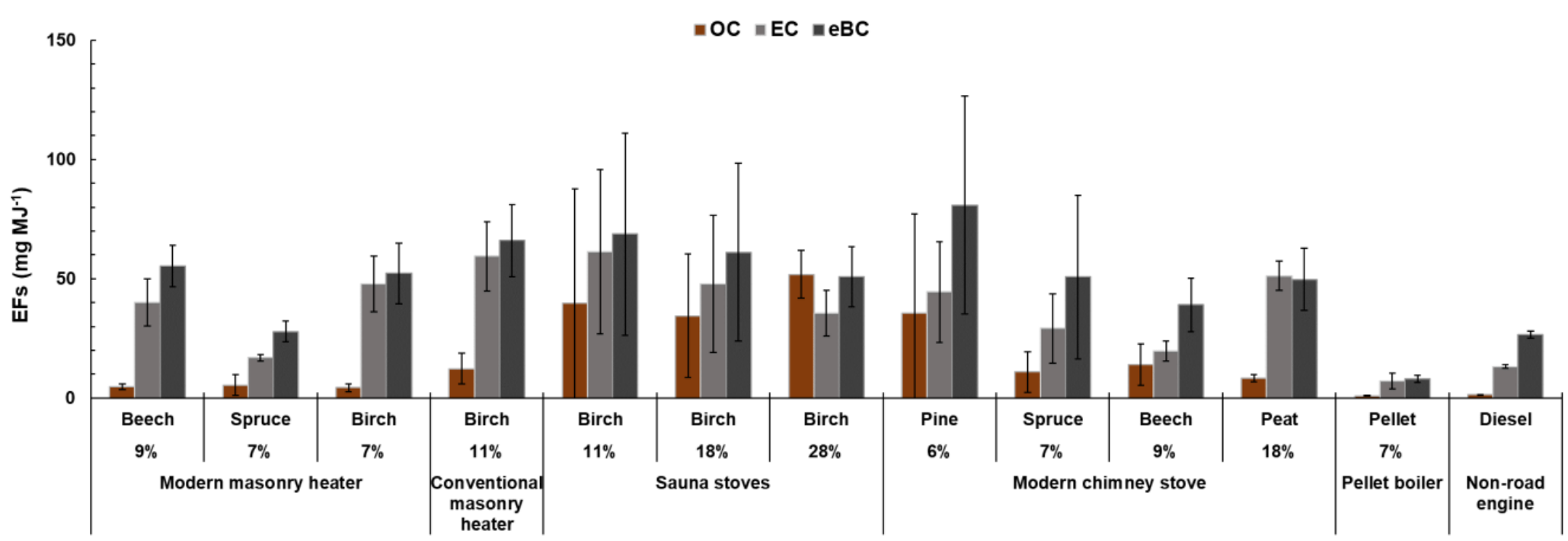

**Figure 1.** Average EFs (mg $MJ^{-1}$ ± standard deviation of the means) for organic carbon (OC), elemental carbon (EC), and equivalent black carbon (eBC) concentrations. The percentages shown indicate the fuel moisture content.

The particle size distributions resulting from wood log combustion were generally characterized by a single dominant mode, with aerodynamic geometric mean diameters (GMDs) ranging from 58 nm (MMH, beech) to 128 nm (SS-18, birch) (Supp. Table S4). For pellets, peat, and diesel, the GMD values were 58 nm, 64 nm, and 44 nm, respectively.

The mean MCEs for wood log combustion varied from 0.94 to 0.98, with the lowest values observed in the SS experiments (Supp. Table S2). These values were notably higher than typical values reported for residential wood combustion in developing countries (Andreae, 2019) but were in the range of European / North American wood log stoves (Tasoglou et al., 2017; Price-Allison et al., 2021). An increase in the fuel moisture content decreased MCE values which is in agreement with the previous studies (e.g., Price-Allison et al., 2021, 2019) (Supp. Figure S6b). For the pellet boiler, the MCE was 0.998 ± 0.002, indicating a nearly complete combustion. Generally, an MCE value less than 0.9 refers to smoldering combustion while higher values correspond to flaming combustion (Reid et al., 2005; Shen et al., 2011). Based on this, all the experiments could be classified as flaming combustion, although our experiments also include the ignition phase and residual char burning phases. There was no observed correlation between the MCE and EC or eBC emissions, but a slight negative correlation ($R^2 = 0.62$, $p = 0.001$) between average OC emissions and MCE for the fuel-appliance combinations (Supp. Figure S7).

### 3.2 eBC absorption and $MAC_{880nm}$

The potential differences in the absorption efficiency of soot emitted from the different combustion sources were investigated by determining the mass absorption cross-section of BC ($MAC_{880nm}$) for each fuel-appliance combination based on the linear regression model between $b_{abs(880nm)}$ and EC mass concentrations. However, it is important to note that since we did not have a reference optical instrument to correct for potential bias in the aerosol multiple scattering coefficient (C) (Kumar et al., 2018; Tasoglou et al., 2017; Olson et al., 2015; Wu et al., 2021), the absolute $MAC_{880nm}$ values should be treated as indicative. Among the wood log combustion appliances, the $MAC_{880nm}$ ranged between 8.41 ± 1.34 $m^2\,g^{-1}$ (CMH, birch) and 15.0 ± 5.49 $m^2\,g^{-1}$ (MCS, beech) (Figure 2b). Interestingly, the average $MAC_{880nm}$ for each wood species with similar moisture content did not differ significantly even when combusted in different appliances. However, the beech logs showed an exception with a $MAC_{880nm}$ of 9.98 ± 4.13 $m^2\,g^{-1}$ in MMH and 14.0 ± 7.78 $m^2\,g^{-1}$ in MCS. Overall, the average $MAC_{880nm}$ values for pine and spruce log combustion were 12.20 ± 2.05 $m^2\,g^{-1}$ and 13.30 ± 2.05 $m^2\,g^{-1}$, respectively, which were higher compared to birch (9.40 ± 1.65 $m^2\,g^{-1}$). The $MAC_{880\,nm}$ for pellet burning was 8.16 ± 2.84 $m^2\,g^{-1}$ and for NrDE 15.6 ± 0.75 $m^2\,g^{-1}$.

Higher $MAC_{880nm}$ values were found for the modern stoves compared to conventional and sauna stoves. This might be a consequence of different nanostructures and the maturity of soot between traditional and advanced appliances. Similar difference has been previously observed for simple and improved cook stoves (Wu et al., 2021; Saliba et al., 2018). We further hypothesized that $MAC_{880nm}$ values may be related to the particle size distribution (Romshoo et al., 2021); however, no correlation was seen between GMD and $MAC_{880nm}$ (Supp. Figure S8).

The $MAC_{880nm}$ increased with increasing fuel moisture content, as observed in the SS experiments (Figure 2b, Supp. Figure S9a). OC/EC ratio showed a moderate correlation with $MAC_{880nm}$ (Supp. Figure S10), suggesting that the increase in $MAC_{880nm}$ might be caused by the increased organic coating enhancing the lensing to the absorbing EC core (Bond et al., 2006; Jacobson, 2000, 2001).

The direct comparison of MACs from different studies is often hindered by differences in instrumentation, as large differences between $MAC_{880nm}$ can be derived even for the same experiment using different BC measurement methods (Healy et al., 2017). Kumar et al. (2018) found beech log combustion soot to have $MAC_{880nm}$ of 4.7 $m^2\,g^{-1}$ based

on Multiwavelength Absorption Analyzer (MWAA) measurements, while Tasoglou et al. (2017) derived $MAC_{532nm}$ of 8.56 $m^2$ $g^{-1}$ and 8.98 $m^2$ $g^{-1}$ for pine and birch bark respectively, using a photoacoustic extinctiometer and refractory BC from Soot Particle Aerosol Mass Spectrometer in their measurements. The aethalometer, on the other hand, measures light attenuation, making our results sensitive to potential discrepancies in the filter multiple scattering. Here, we applied the C values given by the instrument manufacturer for the filter tapes in use. Previous studies have, however, determined C values of 3 for wood log combustion (Kumar et al., 2018) and 2.29–3.45 for ambient air measurements (Yus-Díez et al., 2021; Backman et al., 2017). By using a C value of 3, the MAC factors determined in this work decreased by a factor of ~ 50 % and correspond to the range of 3–8 $m^2$ $g^{-1}$ for wood log combustion (Figure 2b) which is generally in agreement with values reported by Kumar et al. (2018) and Olson et al. (2015). Similarly, using a C value of 3, the $MAC_{880nm}$ of pellet burning (4.27 $m^2$ $g^{-1}$) was comparable to those of the previously studied natural draft pellet-fired stove with AE31 and photoacoustic extinctiometer (PAX532) (Olson et al., 2015). However, even with the C value of 3, the $MAC_{880nm}$ of NrDE (8.16 $m^2$ $g^{-1}$) remains higher compared to previous studies (Olson et al., 2015; Wu et al., 2021). This difference in $MAC_{880nm}$ for diesel engine exhausts may be due to the differences in engine and exhaust after-treatment technologies. Our test engine was a small NrDE (single-person driven lawnmower) without any emission after-treatment technologies, while the previously reported MAC values are predominantly for heavy-duty vehicles equipped with exhaust after-treatment systems, such as diesel oxidation catalyst and exhaust gas recirculation. These after-treatment systems have been previously shown to alter soot nanostructure, which also likely influences the soot optical properties (Malmborg et al., 2017).

### 3.3 Absorption Ångström exponents

The two-wavelength AAE ($AAE_{470/950}$) model is used widely in literature and is currently a standard method applied for the source apportionment of ambient aerosols. Thus, here we focus the discussion on $AAE_{470/950}$. Generally, the $AAE_{470/950}$ and $AAE_{370-950}$ were in line, with a slightly higher $AAE_{470/950}$ compared to the $AAE_{370-950}$. The average $AAE_{470/950}$ for full experiments using moderately dry (7–11% moisture content) wood log combustion varied between 1.06 ± 0.31 (MMH, birch) and 1.26 ± 0.18 (SS-11, birch) (Figure 2a). These values are similar to previously studied fresh emissions from wood combustion (Helin et al., 2021; Kumar et al., 2018; Grieshop et al., 2017; Tasoglou et al., 2017; Martinsson et al., 2015; Saleh et al., 2013) but notably lower than the assumption of AAE = 2, which is the standard used by the aethalometer ambient source apportionment for biomass burning (Sandradewi et al., 2008). The source apportionment method would, in other words, fail to account for the sources of fossil fuel and wood burning in this study. $AAE_{470/950}$ for the NrDE exhaust was 1.02 ± 0.05, which agrees with the general assumption of AAE ~1 for fossil fuel sources. Similar results for diesel engines have been obtained in previous studies irrespective of the engine types or after-treatment technologies used (Helin et al., 2021; Olson et al., 2015).

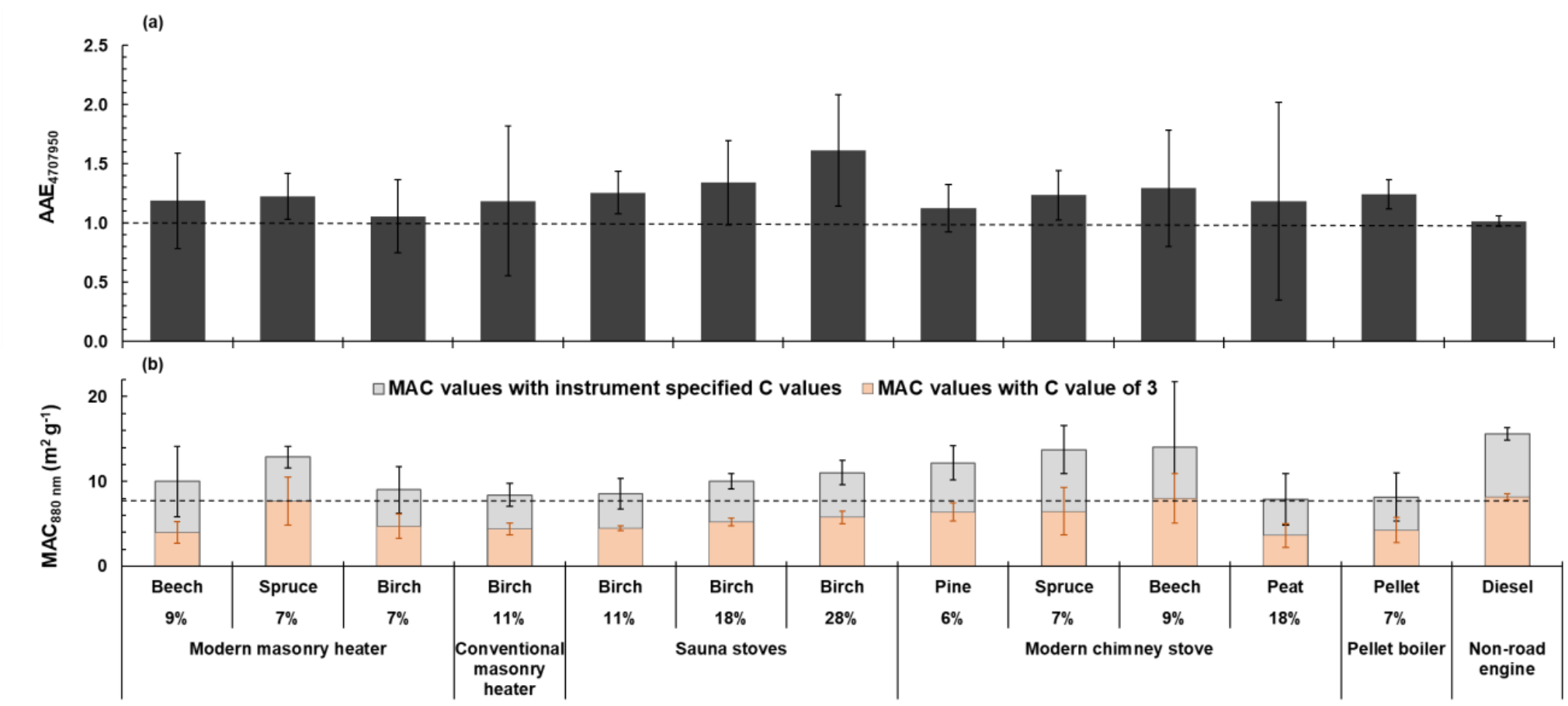

**Figure 2.** Optical properties of aerosols emitted from different combustion processes: average a) $AAE_{470/950}$ and b) $MAC_{880nm}$ from different emission sources, applying the default C values (grey) and with the C value of 3 (orange). The common threshold value of AAE = 1 for pure BC and the default $MAC_{aeth}$ for 880 nm (7.77 $m^2\ g^{-1}$) are shown as dashed lines for reference. Error bars illustrate the standard deviations of the means over whole burn cycles. The percentages shown indicate the fuel moisture content.

The design of the sauna stove had minimal influence on the $AAE_{470/950}$ when dry fuel was used, as shown in Supp. Figure S9b. However, when the fuel moisture content increased from 11 % to 28 %, the $AAE_{470/950}$ changed from 1.26 ± 0.18 to 1.61 ± 0.47 (Figure 2a, Supp. Table S5). This change in $AAE_{470/950}$ can be attributed to differences in the fuel moisture content affecting the combustion conditions in the stove, with higher $AAE_{470/950}$ associated with lower average MCEs (Figure 3b). This suggests more BrC was produced during less efficient combustion conditions. A similar converse relationship has been also previously observed (Zhang et al., 2020; Pokhrel et al., 2016; Tian et al., 2019; Zhang et al., 2021).

$AAE_{470/950}$ of wood combustion exhaust correlated with OC/EC ratio ($R^2 = 0.79$, $p < 0.001$, Figure 3a) due to the BrC constituents in the OC increasing the absorption specifically at the lower wavelengths. Consequently, $BrC_{370\text{-}950}$ showed a very good correlation with $AAE_{470/950}$ ($R^2 = 0.87$, $p < 0.001$) (Supp. Figure S11). In addition to the presence of light-absorbing organic material in the particles, variation in the AAE indicates changes in the extent of lensing of different wavelengths, which is impacted by the morphology, BC core particle size, and coating thickness (Virkkula, 2021; He et al., 2015).

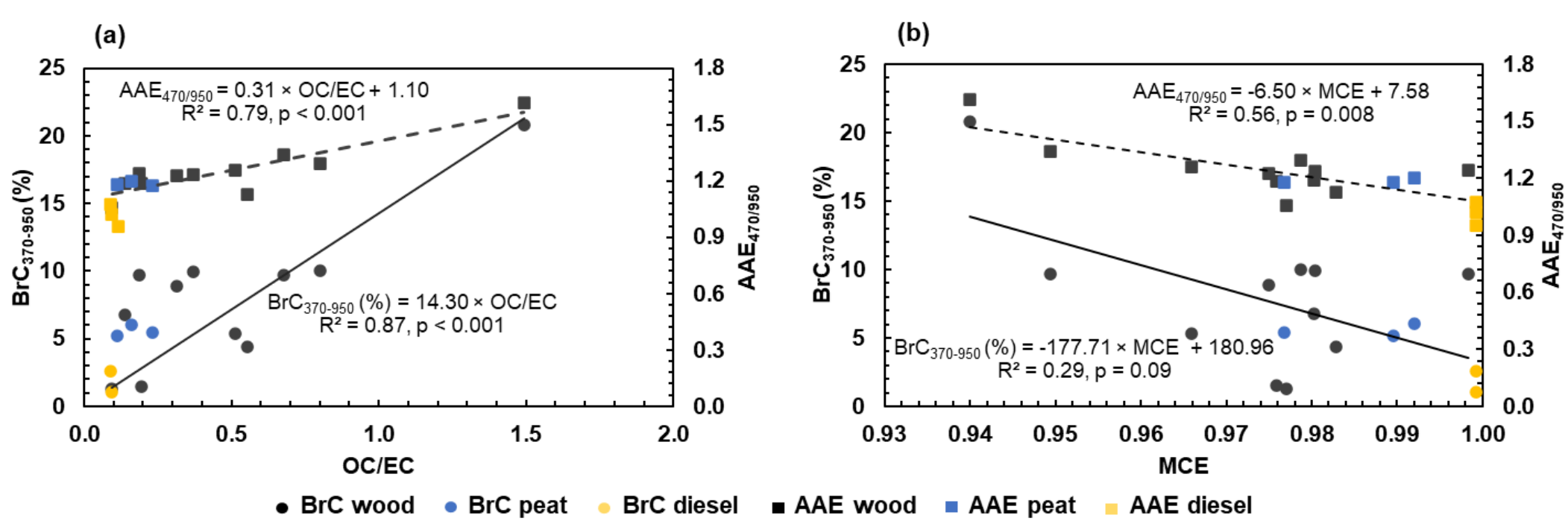

**Figure 3.** Relationships between a) $AAE_{470/950}$ and $BrC_{370\text{-}950}$ (%) vs OC/EC and b) dependence of $AAE_{470/950}$ and $BrC_{370\text{-}950}$ (%) on MCE for individual combustion appliance/fuel combinations. The slopes are calculated for wood combustion, i.e., data for peat and diesel were not considered.

High variability in wood combustion $AAE_{470/950}$ can be observed between the different combustion phases (Figure 4a, b, e, g, and h). The addition of a new batch of wood logs onto the glowing embers led to rapid, low-temperature pyrolysis of volatile organics resulting in high organic aerosol (OA) emissions (Kortelainen et al., 2018). Consequently, the highest $AAE_{470/950}$ (1.5–2.5) was recorded during the ignition phase of each new batch (Figure 4a, b, e). As the flaming phase started, the combustion temperature increased with BC dominating over OA, resulting in a decrease of $AAE_{470/950}$ values to 1.0–1.5 (Figure 4a, b, e, g). Similar observations were seen by Martinsson et al. (2015), with AAEs of exhaust from traditional Scandinavian wood stoves ranging from 1.0–1.2 during flaming phase combustion to 2.5–2.7 during the ignition phase from batch-wise combustion.

Peat had a constant $AAE_{470/950}$ of 1.2–1.3 for the flaming phase with high variability in the ember phase as soon as fuels were consumed before the addition of the next batch (Figure 4f). NrDE use and pellet combustion, on the other hand, had constant $AAE_{470/950}$ values throughout the whole experiment (Figure 4c and d).

The validity of the general assumption of $AAE_{BC}$ = 1 for pure BC particles cannot be directly confirmed with this instrumentation, but it can be reviewed by considering the particles with minimal organic matter content. For example, the particulate exhaust from the birch log combustion in the MMH had a carbonaceous fraction consisting of 91.7 % EC and yielded an $AAE_{470/950}$ of 1.06, relatively closely matching the general assumption of $AAE_{BC}$ = 1. Similarly, diesel engine particles with an EC fraction of 90.8 % had an $AAE_{470/950}$ of 1.02. Further, the validity of the assumption can be examined by extrapolating the correlation between $AAE_{470/950}$ and OC/EC (Fig. 3a). Based on the regression model, RWC soot consisting purely of EC (OC/EC = 0) would have an $AAE_{470/950}$ of 1.1.

In general, our study showed $AAE_{470/950}$ values below 1.4 in fresh exhaust from residential combustion of dry wood. However, ambient aerosols do not only include freshly emitted particles but are a mixture of primary and secondary aerosols. The mixing is both internal (i.e., different chemical compounds are assumed to be mixed in particles for a given particle size), and external (i.e., no physical or chemical interaction occurs between different particle compounds as different chemical compounds are assumed to be present in separate particles). As a result of atmospheric processing, aged emissions may have different $AAE_{470/950}$ values compared to those for freshly produced aerosols. This is due to possible BrC formation or loss from the oxidation and functionalization reactions of organic matter in the atmosphere (Yang et al., 2021; Liu-Kang et al., 2022; Hems and Abbatt, 2018), changes in the lensing by particle coating (Lack and Cappa, 2010; Hems et al., 2021) and other morphological changes in particle structure (Leskinen et al., 2023). Thus, lower $AAE_{470/950}$ values in this study compared to the ambient wood-burning aerosols may be partially explained due to the lack of atmospheric aging of measured emissions.

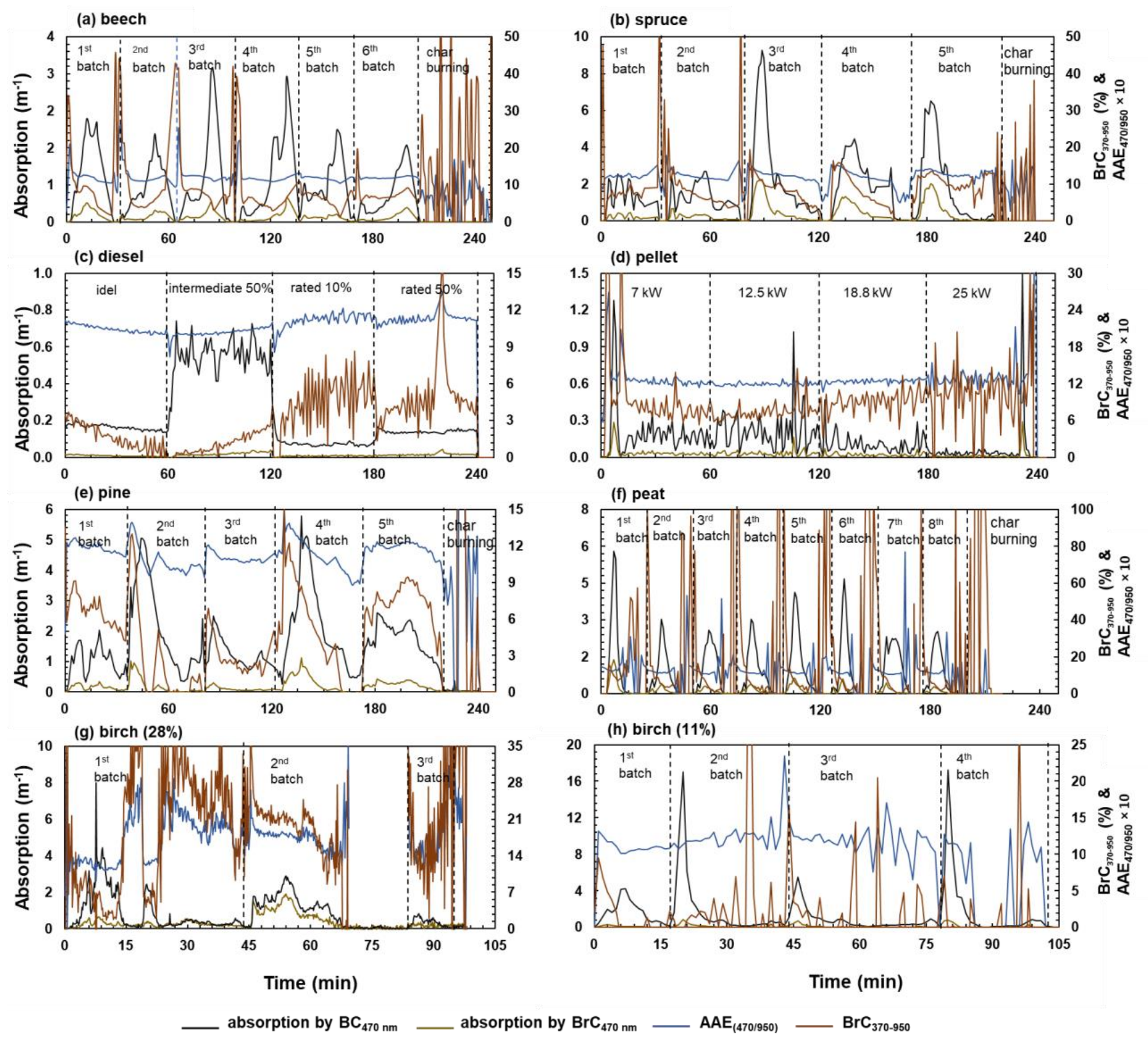

**Figure 4.** Examples of the time series for absorption by BC (black) and BrC (yellow) ($m^{-1}$ in raw exhaust gas concentrations) at 470 nm along with $AAE_{470/950}$ values (blue) and $BrC_{370\text{-}950}$ total absorption (brown) (%) for combustion experiments using different fuel types in following combustion appliances: a) MMH, b) MCS, c) non-road diesel engine, d) PB, e) MCS, f) MCS, g) SS, and h) CMH. In (g), $AAE_{470/950}$ and $BrC_{370\text{-}950}$ are not shown for the ember phase (69–84 min) due to the low PM emission.

### 3.4 Absorption by BrC

The $BrC_{370\text{-}950}$ varied between different combustion events, as illustrated in Figure 4. The total absorption coefficients below the wavelength of 880 nm were higher than those extrapolated for BC at $AAE_{BC}$ = 1, indicating the presence of BrC (Figures 5 and 6) and potentially also the enhanced lensing effect due to the coating of soot particles. For wood log combustion, $BrC_{370\text{-}950}$ ranged from 1.28 % for birch with 7 % moisture content up to 20.8 % for birch with 28 % moisture content (Figure 6). Surprisingly, peat combustion displayed a relatively low $BrC_{370\text{-}950}$ (5.98 %) at 18 % moisture content, presumably because of the prevalent flaming combustion resulting in minimal organic emissions. In the case of NrDE use, $BrC_{370\text{-}950}$ was negligible (0.66 %), consistent with expectations based on prior studies (Sandradewi et al., 2008). This marginal BrC absorption can be attributed to internal mixing, as diesel engines are not anticipated to generate aerosols containing BrC (Malmborg et al., 2021).

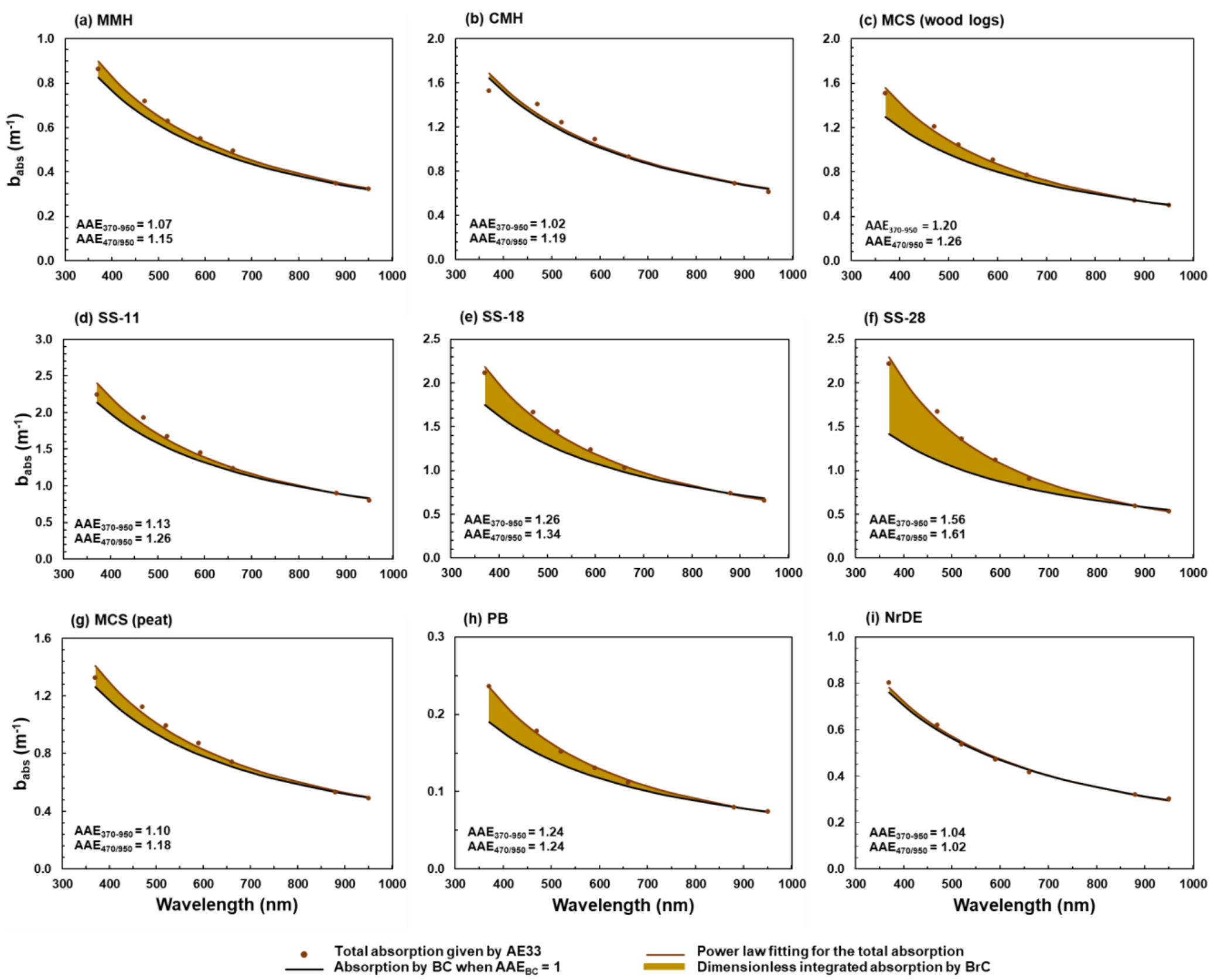

**Figure 5.** Absorption (m$^{-1}$ in the raw exhaust (plume) gas concentration) by $BrC_{370\text{-}950}$ summarized for combustion appliances: a) MMH, b) CMH, c) MCS with wood logs, d) SS-11 with 11 % fuel moisture content, e) SS-18 with 18 % fuel moisture content, f) SS-28 with 28 % fuel moisture content, g) MCS with peat, h) PB, and i) NrDE. The brown markers indicate the total absorption given by AE33, while the brown line represents the power law fitting for the total absorption given by AE33. The black line represents the absorption by BC when $AAE_{BC}$ = 1, and the difference between the total absorption and absorption by BC, shown with a yellowish-brown area, indicates the dimensionless integrated absorption by BrC.

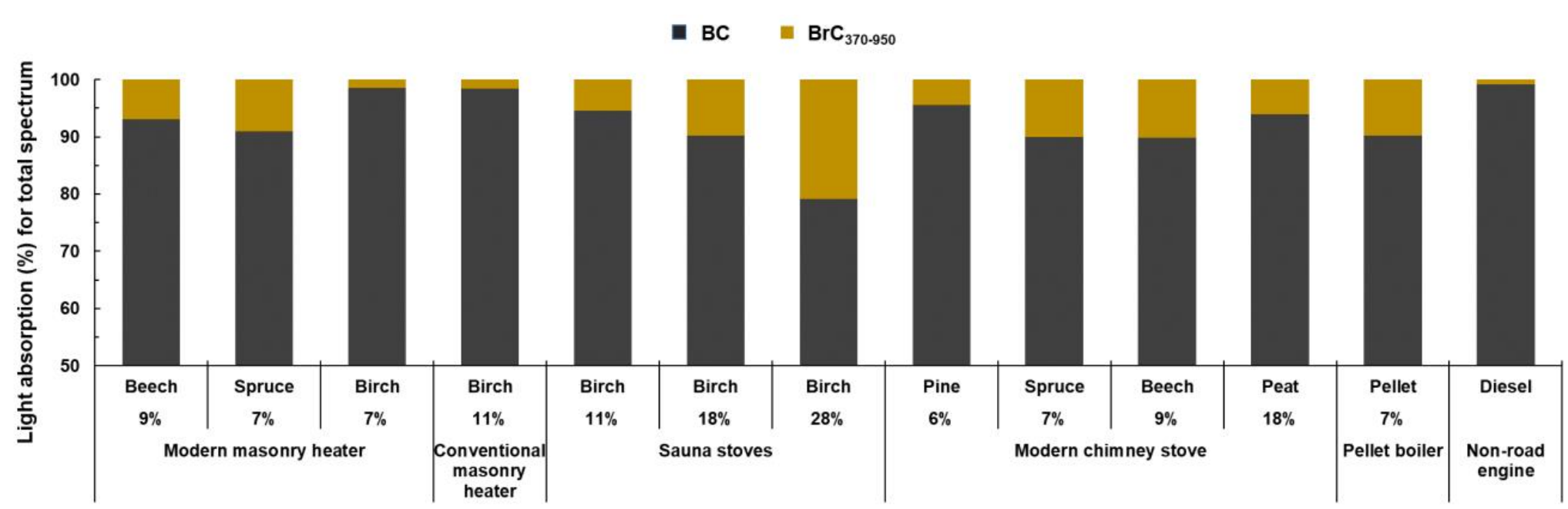

**Figure 6.** Contribution of BC and BrC to the total absorption over the wavelength range 370–950 nm for different fuel types and combustion appliances ($AAE_{BC}$ = 1). The percentages shown indicate the fuel moisture content.

The contribution of BrC to the absorption at 470 nm is higher than at other wavelengths for most of the experiments (Figure 5, Supp. Table S6), resulting in lower $AAE_{370\text{-}950}$ compared to the $AAE_{470/950}$. The average $BrC_{370\text{-}950}$

was approximately 60 % of the contribution of BrC exclusively at 470 nm (Figure 7). Although BrC has a significant presence and absorption capability across a broader spectral range, the evaluation of BrC absorption often focuses on specific wavelengths, such as 370 nm or 470 nm. In a study conducted by Martinsson et al. (2015) with birch log combustion, BrC accounted for 20–75 % of the absorption at lower wavelengths during the ignition phase. The BrC absorption decreased with increasing wavelengths and was reported to account for 45–65 % at 520 nm. Similarly, Pokhrel et al. (2017) reported a wide range for the contribution of BrC to absorption at 405 nm (from 0 % to 92 %) and 532 nm (up to 58 %) for biomass combustion exhausts depending on the utilized segregation method, while Fang et al. (2022) observed BrC absorption ranging from 33 % to 90 % at 370 nm during different biomass burning experiments. The environmental significance of the absorptivity at different wavelengths also varies depending on the spectral abundance in the atmosphere. Thus, future studies should include a description of the BrC's contribution across the visible wavelength in order to comprehensively describe the aerosol's impact on radiative forcing.

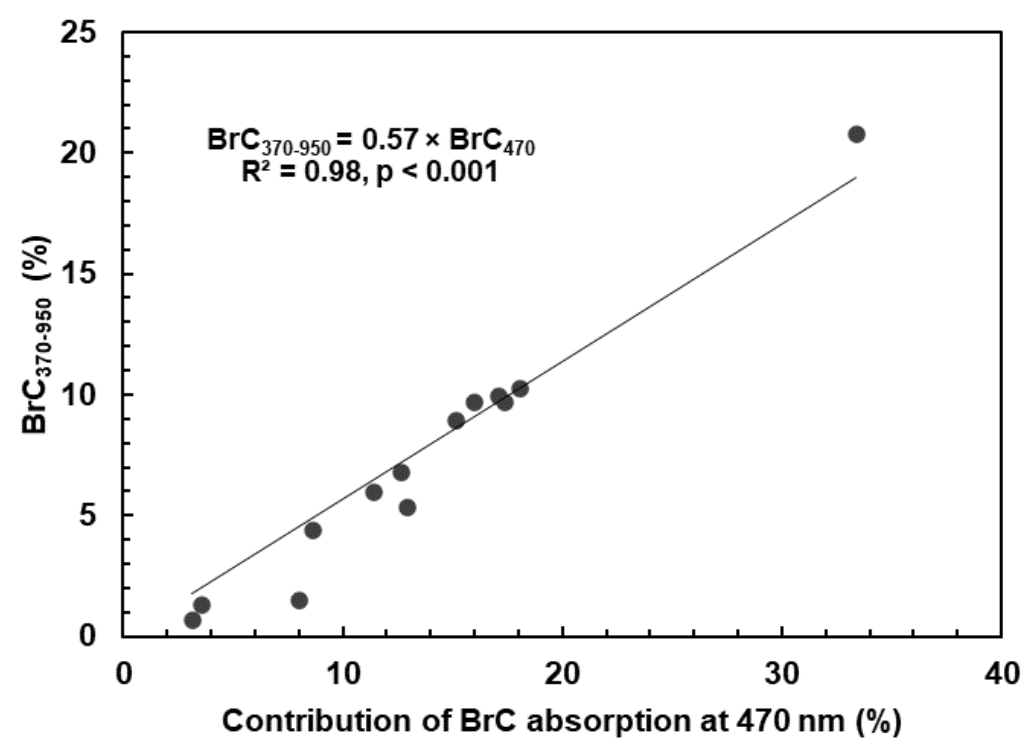

**Figure 7.** Relation between the contribution of BrC to the total absorption ($BrC_{370-950}$) and BrC absorption exclusively at 470 nm.

The contribution of BrC depends on the applied $AAE_{BC}$ values, which are indicative of the spectral dependence of absorption by the BC. The sensitivity of the BrC contribution to the chosen $AAE_{BC}$ value depends on the amount of BrC, as depicted in Supp. Figures S12 and S13. The relative uncertainty associated with the assumed $AAE_{BC}$ increases when the fraction of BrC in the emissions is low. For instance, assuming an $AAE_{BC}$ of 0.9 led to an almost four-fold increase in the contribution of BrC (from 1.28 % to 5.07 %) for dry birch wood and seven-fold (from 0.66 % to 4.47 %) for diesel fuel when compared to the initial shares with an $AAE_{BC}$ of 1.0. In contrast, when an $AAE_{BC}$ of 1.1 was assumed, the absorption was completely dominated by BC for these fuels.

The average $BrC_{370-950}$ of the studied fuel/appliance combinations had a very good correlation with the OC/EC ratio ($R^2 = 0.87$, Fig. 3a). An increase of one unit in the OC/EC ratio corresponded to a 14 % increase in the $BrC_{370-950}$. The correlation of $BrC_{370-950}$ with the OC fraction (Supp. Figure S14) suggests that the fraction of absorption by BrC can become a major contributor to UV and near-UV absorption if a significant amount of OC is present in the exhaust emission. Also, the MCE had a slight relationship with the BrC absorption, the overall trend showing that lower MCE values correspond to higher $BrC_{370-950}$, which is in agreement with previous findings concerning open biomass burning (Pokhrel et al., 2017; Li et al., 2019a).

The $MAC_{OC,470nm}$ had high variability ($24.1 \pm 6.44$ m$^2$ g$^{-1}$ to $2.39 \pm 5.86$ m$^2$ g$^{-1}$) for the different fuel/appliance combinations, with no clear trend between the $MAC_{OC,470nm}$ and the amount of OC. This variation may be due to differences in the chemical composition of the organic fraction or the extent of the lensing effect with varying coating

amounts, and determination of the exact $MAC_{OC}$s would require further investigation of the pure organic fraction from different combustion conditions.

## Conclusions

In this study, we characterized the emissions of BrC originating from various wood combustion appliances in Northern Europe based on the wavelength-dependent aerosol absorption data from a 7-wavelength aethalometer and thermal-optical carbon analysis of particle filter samples. By utilizing the information from all seven wavelengths of the aethalometer, we derived an integrated parameter to account for the total BrC absorption across the visible light spectrum. The $BrC_{370\text{-}950}$ varied greatly (ranging from 1.28 % to 20.77 %) for wood log combustion events and was primarily influenced by fuel moisture content and modified combustion efficiency. Notably, the type of combustion appliance and wood log species had only a slight effect on the BrC emissions. Higher fuel moisture content led to decreased combustion efficiency, resulting in higher $BrC_{370\text{-}950}$ contributions in the combustion emissions. Furthermore, correlations between the OC/EC ratios and $BrC_{370\text{-}950}$ demonstrated that a one-unit increase in OC/EC would correspond to an approximately 14 % increase in $BrC_{370\text{-}950}$. These findings confirm that BrC can be a significant contributor to UV and near-UV light absorption in wood stove emissions from Northern Europe when OC/EC ratios are high, and may offer a useful parametrization for assessing the BrC emissions from different RWC designs.

The study also reviewed the commonly used assumption of an AAE of 2 for source apportionment of wood-burning emissions. The $AAE_{470/950}$ values obtained in this study ranged from 1.06 to 1.61 for wood log combustion, suggesting that the assumed value of AAE = 2 might be too high for Northern European RWC emissions. However, it should be noted that the presented $AAE_{470/950}$ values were determined for fresh aerosols, and atmospheric transformation is likely to influence these values. By extrapolating the correlation between $AAE_{470/950}$ and OC/EC, it was found that pure RWC black carbon particles would exhibit an $AAE_{470/950}$ close to unity, which aligns with the general assumption used to differentiate the light absorption caused by black carbon and brown carbon. Additionally, exhaust from a non-road diesel engine was found to have an $AAE_{470/950}$ of 1.02, matching well the assumptions used in source apportionment. During the wood log combustion process, both $BrC_{370\text{-}950}$ contribution and $AAE_{470/950}$ varied substantially, with the ignition phase exhibiting higher BrC contributions compared to the flaming phase. This disparity can be attributed to the introduction of a new batch of wood logs onto the glowing embers, resulting in rapid, low-temperature pyrolysis of volatile organics and subsequent high emissions of organic aerosols.

Overall, this study provides valuable data for the estimation of BrC emissions from Northern European RWC. Additionally, the findings can aid in the estimation of radiative forcing by BrC using multi-wavelength aethalometer data in air quality monitoring networks. However, further research concerning different combustion conditions and fuel qualities of RWC, especially high moisture content fuels, and low combustion efficiencies, is needed to comprehensively evaluate the role of BrC in particle light absorption for RWC in other regions.

## Data availability

Further experimental data will be provided upon request to the corresponding authors.

525 **Supporting Information**

Supporting information contains additional information on the experimental conditions, a description of BrC uncertainty calculation, and other experimental results to support the findings of this study.

**Author contributions**

OS, AK, RZ, and JT supervised and acquired funding for the study. MI, PYP, OS, GJ, MK, HS, SV, JL, MS, JT, SB, LK, and AH designed and performed the experiments. SB, AH, OS, SV, MK, AV, and HS analyzed and evaluated the data. SB, AH, and OS wrote the paper with input and approval from all authors.

**Competing interests**: The authors declare that they have no conflict of interest.

**Acknowledgments**

Funding: This work was supported by the Finnish Cultural Foundation, North Karelia and North Savo Funds (AERO-LCA project, grant nos. 55201441 and 65202068), the Research Council of Finland (BBrCAC project, grant no. 341597), the "KIUAS-2" project, and the Helmholtz Virtual Institute of Complex Molecular Systems in Environmental Health, Aerosols and Health (HICE) funded by the Initiative and Networking Fund of the Helmholtz Association (HGF, Germany). ChatGPT-3, an open AI tool, was consulted for paraphrasing the conclusions section of this manuscript.

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
