# Peer review of "Contribution of brown carbon on light absorption in emissions of European residential biomass combustion appliances"

_EGUsphere, 2023_

## Author Comment (AC2)

Response to Reviewer 2's comments

This paper presents optical properties of fresh smoke measured with a 7-wavelength aethalometer (AE33) from various wood burning devices using different fuels. Emissions from a diesel engine are also included. Additional measurements include the smoke OC, EC concentrations, CO and CO2 to determine the MCE (and NOx). A significant amount of data is presented. The results are generally as expected, when the fuel is wet (or when fresh wood is placed on the fire) the combustion is more smoldering, MCE drops and OC/EC and BrC increases. Many of the graphs need more explanation or improved clarity. More discussion on the effect of coatings over BC on the overall AAE is needed. Possible effects of tar balls or dark BrC is never discussed. Overall, the paper is interesting and the detailed analysis adds to the data base on residential wood heating emissions, but improvements could be made.

We thank Reviewer 2's positive comments and constructive suggestions for improving the manuscript. The responses to the comments are addressed in blue text. The line number refers to the lines in the revised manuscript.

In the revised version, we have made updates to the figures to enhance readability. Additionally, we have incorporated more detailed explanations directly within the text, aiming to provide a clearer and more comprehensive understanding of the content. The other explanations regarding the effect of coatings over BC on the overall AAE and the possible effects of tar balls are discussed below within the specific comments.

Specific Comments;

Line 41-42 & 46. What is small scale residential combustion? Is it different than low-T biomass/biofuel combustion? Is it RWC?

In lines 41-42, it was an alternative wording to the residential wood combustion (RWC) and has now been changed to RWC for consistency. In line 46, the term "small-scale combustion" is used instead of RWC in order to encompass also the possible contribution of residential combustion of other solid fuels, such as coal, to European air pollution.

Line 250, what is evaporation heat of water. I think the term is water latent heat of vaporization.

We have revised the terminology "evaporation heat of water" to "latent heat of vaporization of water," as indicated in the updated text (line 252).

It is unfortunate that lambda has two separate meanings in this paper. Why not use a different variable for the air to fuel ratio (line 246)?

We appreciate your acknowledgment and the issue regarding the use of the same symbol (*lambda*, $\lambda$) for two different variables has been addressed. In the updated version, the symbol for the air-to-fuel ratio has been modified to *alpha* ($\alpha$), as reflected in line 247 and equations 7, 8, and 9 (lines 246, 254, and 255).

Line 260, what is the meaning of the ' in the denominator in equation 12?

This was a comma (,) which is now removed from equation 12 in the updated version (line 260).

Line 178 and line 312 and table S4, are the particle size distribution data for number or mass distributions? Please specify throughout, but it should be stated clearly in line 178 when discussing the electric low pressure impactor.

The manuscript has been updated to specify that the size particle distribution data mentioned in this study refers to number size distributions. This information is now stated in lines 178 and 312, as well as in the list of abbreviations and the Table S4.

Line 332, what exactly is the meaning of indicative? Is the meaning that the magnitude could be biased relative to the real value (whatever that is), but the comparisons between stoves-fuels are ok (ie, like the comparisons shown in Fig 2b)?

Indeed, the term "indicative" (line 333) in this context emphasizes that the $MAC_{880nm}$ presented in this manuscript should not be directly compared with those from other studies utilizing different instrumentation. The variations in measurement techniques and devices require caution in making such direct comparisons. However, the comparability within the manuscript assessing stoves and fuels remains valid, as these comparisons are internally consistent, relying on uniform methods and instruments (aethalometer AE33 measurements) across the specified experiments.

Could the variability in MAC880 also be due to the presence of tar balls or dark BrC which does absorb at the highest wavelengths? The possible role of these forms of BrC should be discussed.

This is a relevant question, as it has been shown that tar balls are emitted from various sources, including biofuel and biomass combustion (e.g., Pósfai et al., 2004), and that they absorb light in the full spectrum from UV to IR wavelengths (Corbin et al., 2019). However, in the present study, we had no means of estimating the contribution of absorption by tar balls to any wavelengths. We have previously carefully characterized the particle morphology for the emissions from the modern masonry heater (Leskinen et al., 2014) and the modern chimney stove (Leskinen et al., 2023) used in this study using electron microscopy and the APM-SMPS method. There were no indications of tar ball -type morphologies in particulate emissions from either of those devices in similar combustion events as covered by this study. However, it is still possible that tar balls or other "dark-BrC" -like organic matter, influencing light absorption in the highest wavelengths, have existed in some of the studied aerosols, particularly those with high OC/EC.

We have now added the following discussion in the results & discussion (line 348 onwards): "$MAC_{880nm}$ might also be enhanced by the presence of organic matter absorbing light also at the higher wavelengths. Specifically, tar ball like morphologies in biomass combustion particles have been associated with a wide absorption wavelength range (Hoffer et al., 2016). In previous studies, tar ball like morphologies were not observed in the emissions of the wood-fired modern chimney stove (Leskinen et al., 2023) or the masonry heater (Leskinen et al., 2014). However, less efficient wood combustion conditions may promote the formation of tar ball like particles (Pósfai et al., 2004)."

References:

Corbin, J. C., Czech, H., Massabò, D., de Mongeot, F. B., Jakobi, G., Liu, F., Lobo, P., Mennucci, C., Mensah, A. A., Orasche, J., Pieber, S. M., Prévôt, A. S. H., Stengel, B., Tay, L. L., Zanatta, M., Zimmermann, R., El Haddad, I., and Gysel, M.: Infrared-absorbing carbonaceous tar can dominate light absorption by marine-engine exhaust, npj Clim. Atmos. Sci., 2, 10.1038/s41612-019-0069-5, 2019.

Leskinen, J., Ihalainen, M., Torvela, T., Kortelainen, M., Lamberg, H., Tiitta, P., Jakobi, G., Grigonyte, J., Joutsensaari, J., Sippula, O., Tissari, J., Virtanen, A., Zimmermann, R., and Jokiniemi, J.: Effective density and morphology of particles emitted from small-scale combustion of various wood fuels, Environ. Sci. Technol., 48, 13298–13306, 10.1021/es502214a, 2014.

Leskinen, J., Hartikainen, A., Väätäinen, S., Ihalainen, M., Virkkula, A., Mesceriakovas, A., Tiitta, P., Miettinen, M., Lamberg, H., Czech, H., Yli-Pirilä, P., Tissari, J., Jakobi, G., Zimmermann, R., and Sippula, O.: Photochemical Aging Induces Changes in the Effective Densities, Morphologies, and Optical Properties of Combustion Aerosol Particles, Environ. Sci. Technol., 57, 5137–5148, 10.1021/acs.est.2c04151, 2023.

Pósfai, M., Gelencsér, A., Simonics, R., Arató, K., Li, J., Hobbs, P. V., and Buseck, P. R.: Atmospheric tar balls: Particles from biomass and biofuel burning, J. Geophys. Res. Atmos., 109, 10.1029/2003jd004169, 2004.

Line 343, GMD of what, number or mass? Isn't mass distribution what matters here?

We refer to the particle number size distributions. This has now been clarified in the text (lines 178, 312, and 343) and the list of abbreviations. The densities were not measured in the experiments, so mass mean diameters cannot be calculated from the data. A reasonable substitute is the volume mean diameter (VMD). For lognormal size distributions, VMD can be calculated using Hatch-Choate equations from VMD = $GMDexp(3ln^2GSD)$ where GSD is the geometric standard deviation of the number size distribution (Hinds, 1999; Baron and Willeke, 2001). Assuming lognormal number size distributions, this equation can be used for estimating the VMD for each of the experiments, with the GMD and GSD results shown in Table S4.

On line 344 it is stated that no correlation was seen between GMD and MAC. The GSDs of the size distributions are approximately $2.0 \pm 0.1$ which means that $VMD \approx 4.2GMD$. In other words, the mean diameter plainly shifts towards larger sizes, and the correlation between the MAC and VMD would be equally poor for VMD as it is with GMD. So, the statement on line 344 would be the same.

Another point about the reviewer's question was that is it reasonable to look for correlations between MAC and number-based GMD. When modeling light absorption by particles, the absorption coefficient babs is calculated as an integral over a number size distribution or in real, discrete size distributions from the sum: babs = sum(Qa(Dp)*((pi*Dp^2)/4)*dN(Dp)/dDp)dDp where Qa(Dp) is the size-dependent absorption efficiency, Dp the particle diameter and dN(Dp) the number of particles in the size bin dDp. Since the MAC of EC is calculated from MAC = babs/EC, where EC is the mass concentration of EC, it is obvious that the number size distribution is an important property influencing MAC values.

References:

Hinds, W. C.: Aerosol Technology – Properties, Behavior, and Measurement of Airborne Particles, John Wiley & Sons, INC., New York, 1999.

Baron, P. A. and Willeke, K.: Aerosol Measurement – Principles, Techniques, and Applications, John Wiley & Sons, INC., New York, 2001.

Fig 2b is not clear. What is the meaning of the overall bar height? Eg, is the MAC with C=3 the sum of the gray and red bars? Needs more explanation.

In Figure 2b, the vertical bars correspond to the $MAC_{880nm}$. Specifically, the height of the grey bar represents the $MAC_{880nm}$ calculated using the default values in the aethalometer (default C value), while the height of the orange bar illustrates the $MAC_{880nm}$ when the C value is adjusted to 3. To address potential confusion arising from the overlapping bar plots, we have revised Figure 2b in the updated version. The modified figure now features the $MAC_{880nm}$ with C = 3 depicted as orange dots rather than bars, aiming to enhance clarity in the visual presentation of the data.

In Fig 3a the AAE vs OC/EC data looks more correlate then BrC vs OC/EC, opposite to the regression (r2) results given in the plot.

Acknowledging the identified mistake in the equation presented in Figure 3a, we confirm that this error has been rectified in the updated version of Figure 3a and line 491 (when referring to the $R^2$ value in Figure 3a). The corrected equation now accurately aligns with the intended representation.

Line 396-399. Is this statement correct, ie does lensing change the AAE to a value different than that of pure BC and is this statement supported by references Virkkula et al and He et al. For example, does a clear

coating (could be OA, or other species) on BC result in an AAE different from the AAE of pure BC? If the coating is OA containing chromophores, is the AAE different from pure BC AAE only due to those chromophores absorbing light (the BC core has no effect) or does the coating-BC combination change the overall AAE? This needs clarification, possible by adding more details.

This argument is made in many places, so the question also applies to line 429, line 441 and line 489.

Yes, the statement is correct. The lensing effect indeed does change the AAE values compared to pure BC. This has been shown in several studies, e.g., Gyawali et al. (2009), Lack and Cappa (2010), Lack and Langridge (2013), and more recently by Virkkula (2021) and Luo et al. (2023). Already Gyawali et al. (2009) showed that if the absorption by the coating is wavelength dependent, the AAE is different from the AAE of pure BC, and consequently, this effect is a combination of lensing and absorption by the coating and that these effects vary as a function of core size and coating ratio.

We have added a few more references to support our statement in the paper (line 406).

The new citations:

Gyawali, M., Arnott, W. P., Lewis, K., Moosmüller, H., and Moosmüller, M.: Atmospheric Chemistry and Physics In situ aerosol optics in Reno, NV, USA during and after the summer 2008 California wildfires and the influence of absorbing and non-absorbing organic coatings on spectral light absorption, Atmos. Chem. Phys, 9, 8007–8015, 2009.

Luo, J., Li, Z., Qiu, J., Zhang, Y., Fan, C., Li, L., Wu, H., Zhou, P., Li, K., and Zhang, Q.: The Simulated Source Apportionment of Light Absorbing Aerosols: Effects of Microphysical Properties of Partially-Coated Black Carbon, J. Geophys. Res. Atmos., 128, 10.1029/2022JD037291, 2023.

Fig 4 is not very useful, maybe better to extract the important information and move it to the supplement.

Acknowledging the suggestion, we recognize that Figure 4 could be more suitably placed in the supplement. Accordingly, in the updated version of the manuscript, Figure 4 has been relocated to Figure S12 in the supplementary material.

Line 458 and 459. This line seems incorrect, or at least the meaning is not clear. The contribution of BrC (ie, b(abs) just due to BrC) at 370 nm is always higher than at 470 nm. The issue here is more related to what wavelength ranges are used for the fit. I believe what is being noted here is that when using the lowest wavelength, the fit produces a lower AAE.

The measured relative contribution of BrC to the absorption at 370 nm was in fact often lower than at 470 nm, as evident in Figure 4 and Table S6. To avoid misinterpretations, we have added the word "relative" on line 464. Our results deviate from the anticipated trend where lower wavelengths would typically exhibit a higher contribution of BrC compared to higher wavelengths. Contrary to this expectation, our observations indicate that the BrC contribution is most pronounced at 470 nm for most experiments. Consequently, the AAE fit over the range of 370-950 nm ($AAE_{370-950}$) using all the seven aethalometer wavelengths tends to be lower compared to the AAE calculated using only the two wavelengths ($AAE_{470/950}$).

---

## Author Comment (AC3)

Response to Reviewer 1's comments

This manuscript by Basnet et al. reports results of emission factors and light-absorption properties of carbonaceous aerosol emissions from residential biomass combustion appliances. The experiments involved an extensive set of fuels (7) and appliances (15). The major measurements involved (i) offline thermal-optical analysis to determine emission factors of elemental carbon (EC) and organic carbon (OC), (ii) online measurements of light-absorption at 7 wavelengths using an aethalometer, and (iii) online measurements of size distributions using a low-pressure impactor. The major analysis involved apportioning absorption to either black carbon (BC) or brown carbon (BrC) based on the assumptions that (i) only BC absorbs at 880 nm and (ii) BC absorptions exhibits a wavelength dependence with AAE = 1. The results show variable contributions to absorption by BC and BrC, with fuel moisture content playing an important role.

We thank Reviewer 1's positive comments and constructive suggestions for improving the manuscript. The responses to the comments are addressed in blue text. The line number refers to the lines in the revised manuscript.

Major comments:

1) There are a lot of previous studies that quantified BrC and BC absorption from residential wood burning. It is not clear if/how this study provides any new insights beyond what is already in the literature. For this paper to be suitable for publication in ACP, it needs to clearly identify the new knowledge generated from the experiments. Given the large data set in this study, there is probably potential for deriving new useful knowledge. However, this is not clear in the current version of the paper. A couple of examples of how the paper could potentially highlight new/important/useful results:

While numerous investigations have quantified the absorption of BrC and BC arising from residential wood burning, the majority of these investigations primarily concentrate on assessing ambient air quality for which the residential wood burning emissions occur as mixed with other air pollutants. Thus, there has been a lack of measurement data on wavelength-dependent absorption for residential wood combustion emissions without any mixing factors (such as other pollutants, photochemical/dark atmospheric transformation, etc.). There has been specifically limited information about BrC emissions and light absorption wavelength dependency from European residential wood combustion appliances. As aethalometers are widely used for, e.g., air quality monitoring, our extensive dataset of the wavelength-dependent absorptions of emissions originating from European residential wood combustion can be used for improved source apportionment.

We thank the reviewer for these suggestions, which we in the following respond to point-by-point.

1.1) Discuss how emission factors from European residential combustion is currently quantified in emission inventories. Can the results obtained here (Figure 1) help improve these emission inventories?

Figure 1 illustrates the emission factors for organic carbon and elemental carbon, derived from the offline thermal-optical analysis along with the equivalent black carbon data obtained from the aethalometer. These emission components are widely covered in emission inventories. However, this study provides useful emission factors for the latest modern northern European appliances. Further, BrC is presently absent from the European emission inventories. The results of this paper enable the inclusion of BrC-induced absorption in emission inventory assessments. The contribution of BrC to the total light absorption for the studied wood combustion appliances is shown in Figure 5 (previous Figure 6).

We have improved the manuscript by adding discussion on this to the introduction (lines 49-55) and conclusions (lines 528-530) sections:

page 3, lines 49-55: "The accurate quantifying of the amount and impacts of the absorbing aerosols emitted from RWC is challenged by the gaps in knowledge regarding the particle optical properties and potential variance in emission factors (EFs). However, only a few RWC appliance types and fuels have been studied

(e.g., Fang et al., 2022; Martinsson et al., 2015; Zhang et al., 2020; Saleh et al., 2013; Kumar et al., 2018; Saliba et al., 2018; Olsen et al., 2020; Li et al., 2022; Leskinen et al., 2023; Tissari et al., 2019; Hartikainen et al., 2020; Savolahti et al., 2016, 2019; Sun et al., 2021), which causes uncertainties in assessing the direct radiative forcing of RWC emissions. Inclusion of BrC in emission inventories is further constrained by the lack of characterization and quantification of BrC emissions."

page 18, lines 528-530: "In addition to providing BC emission factors for the modern RWC appliances, these findings can also aid in integrating the BrC-induced absorption in emission inventory assessments using the multi-wavelength aethalometer data from air quality monitoring networks."

1.2)    The paper mentions that moisture content is more important than the type of appliance in dictating BrC emissions. However, the results (Figures) are not formulated in a way to make use of this finding, or to clearly show that this assertion is valid to begin with. I think that the paper can possibly be restructured to focus on the effect of moisture content versus appliance type. In order to do that, the paper needs to establish the significance of the ranges of moisture contents used in the experiments: Is this variability typical in residential appliances? Data (e.g. BrC absorption) needs to be plotted as a function of moisture content to actually show that moisture content is indeed important and that the data does not cluster based on appliance type.

Fuel moisture content may indeed be one of the factors influencing the brown carbon content in the exhaust. Unfortunately, the impact of fuel moisture content was only studied in detail for two of the sauna stoves, and restructuring around moisture content might conceal other influencing factors. In fact, fuel moisture content alone does not correlate well with BrC contribution. We find the best correlation between particulate OC/EC and BrC contribution, as depicted in Figure 3. Both OC/EC and BrC contribution seems to be influenced by several factors, including fuel moisture, appliance type, wood fuel, and modified combustion efficiency. We have revised the discussion and conclusions texts in the manuscript to better reflect this complexity (i.e., fuel moisture is not a single major parameter influencing the BrC contribution).

We have revised lines 22-23 in the abstract: "Additionally, $BrC_{370-950}$ was clearly influenced by the fuel moisture content and the combustion efficiency, while the effect of combustion appliance type was less prominent."

A discussion of the influencing factors was added on lines 507-510. "The $BrC_{370-950}$ varied greatly (ranging from 1.28 % to 20.8 %) for wood log combustion events and was primarily influenced by fuel moisture content and modified combustion efficiency but also by the combustion appliance type. The highest $BrC_{370-950}$ contributions were observed for the fuels with the highest fuel moisture contents due to the decreased combustion efficiency."

2)    The figures are often hard to follow and are not discussed well in the text. For example:

We have updated the figures and the discussion on their implications in the accompanying text. The changes are described in the following answers.

2.1)    Most of the details in Figure 4 are not discussed in the text. Why were these specific 8 experiments chosen?

Figure 4 provides a comprehensive overview of the temporal changes in total absorption by BrC and $AAE_{470/950}$ during the progression of the experiment. Given the diverse set of experimental data involving various combustion appliances, the selection of these eight figures aimed to encompass at least one experiment from each appliance.

The updated version of this manuscript has moved this figure to the supplementary material (Figure S12, previously Figure 4) while explaining it in the main text (line 410). "The temporal diversity in the $AAE_{470/950}$

is illustrated for all the different fuel and combustion appliance combinations used in this study in the exemplary time series of data in Figure S12."

2.2)    It is not clear what the purpose of Figure 5 is. Figure 5, which has 9 panels, is referred to only once in the text, rather in passing.

Figure 4 (previously Figure 5) summarizes the key findings of this manuscript by illustrating the total BrC absorption within the 370-950 nm range. This is presented as the difference between the overall absorption provided by the aethalometer and the absorption by BC when $AAE_{BC} = 1$.

We recognize the need for detailed referencing of figures in the text and we have provided more thorough explanations in the updated version of the manuscript (line 444 onwards). "Figure 4 summarizes the total absorption coefficients by BrC for the different combustion appliances, with the absorption by BrC in MMH and MCS (Figure 4a and 4c respectively) given as the combinations of the different fuel types (beech, spruce, and birch for MMH, and pine, spruce, and beech for MCS). The average $BrC_{370-950}$ is given for all the individual fuel-appliance combinations in Figure 5."

2.3)    What does Figure 7 signify? And what are the data points?

The data points in Figure 6 (previously Figure 7) present the contributions of BrC to the absorption at 470 nm, which is a commonly used single wavelength for BrC detection, versus the contribution of BrC over the total wavelength range of 370–950 nm. The values are averages for each fuel- and appliance-type combination. This is now clarified in the figure title (line 481). Given that a majority of studies on BrC absorption tend to concentrate on a specific wavelength, such as 370 nm or 470 nm, Figure 6 (previous Figure 7) concisely demonstrates the relation of BrC contribution if considering the total absorption instead of only the commonly used wavelength (470nm) to detect BrC. This observation highlights the importance of comprehensively assessing BrC absorption across a broader spectrum. This is discussed in the text from line 466 onwards.

The Figure 6 title is changed to "The average contributions of BrC to the absorption at 470 nm versus the total contribution over the wavelength range of 370nm–950 nm for all the fuel and appliance type combinations."

3)      What is the physical significance of the dimensionless integrated absorption? If the goal is to quantify the overall contribution to absorption in the atmosphere, the integration has to be performed with respect to the wavelength-dependent intensity of solar radiation. Otherwise, the integration artificially skews the contribution to absorption to shorter wavelengths (because AAE > 0). Usually, experimental results provide wavelength-dependent optical properties, with the understanding that those can be used within radiative transfer calculations that account for the wavelength-dependent solar spectrum. If the authors wish to estimate the integrated contribution to radiative effect by BC and BrC, they could perform 'simple forcing efficiency' calculations (e.g. Chen, Y. & Bond, T. C. Light absorption by organic carbon from wood combustion. Atmos. Chem. Phys. 10, 1773–1787 (2010)).

The dimensionless integrated absorption (DIA) serves as a straightforward parameter aimed at estimating the contribution of BrC to light absorption. This concept revolves around the utilization of a singular parameter to characterize light absorption across the entire measured wavelength interval, as opposed to calculating it individually for each measured wavelength. The adoption of DIA streamlines the assessment of BrC's impact on light absorption, simplifying the analytical approach by condensing the information into a comprehensive parameter. A similar approach has been used previously (e.g., Massabó et al., 2015) for the quantification of the contribution of BrC to the total absorption within the visible wavelength range.

Unfortunately, we cannot reliably perform the proposed 'simple forcing efficiency' (SFE) calculations. The equation for the SFE includes both the total scattering coefficient and the backscattering coefficient. Usually, these are measured with an integrating nephelometer, which was not available for our experiments. Particle

number size distributions could in principle be used for calculating both total and backscattering coefficients. However, our particle size distributions were measured using an ELPI, which gives the distributions as the function of aerodynamic diameter with a relatively low size resolution. These size distribution data are not good enough for calculating the SFE reliably enough. The need for the inclusion of scattering measurements is now also stated in line 480.

Instead of the SFE, we enhanced the estimation on the atmospheric importance of the BrC across the solar spectra by weighting the absorptions by BC and BrC by the standard solar spectra (ASTM G173-03). Due to the parabola-like shape of the standard spectra over the considered wavelength range, the resulting estimate is very close to the reported $BrC_{370-950}$ percentages.

This is now also noted in the manuscript, lines 475-479: "The relative contribution of BrC to the absorptivity in the visible wavelength range remains close to that of $BrC_{370-950}$ also if the absorptions illustrated in Fig. 4 are weighted by the standard solar irradiance spectra (ASTM G173-03; Figure S13). However, in order to comprehensively describe the brown carbon's impact on radiative forcing, future studies should include a description of the BrC's contribution to the absorption across the visible wavelength range as well as the light scattering by the aerosol."

Reference:

Massabò, D., Caponi, L., Bernardoni, V., Bove, M. C., Brotto, P., Calzolai, G., Cassola, F., Chiari, M., Fedi, M. E., Fermo, P., Giannoni, M., Lucarelli, F., Nava, S., Piazzalunga, A., Valli, G., Vecchi, R., and Prati, P.: Multi-wavelength optical determination of black and brown carbon in atmospheric aerosols, Atmos. Environ., 108, 1–12, 10.1016/J.ATMOSENV.2015.02.058, 2015.

Minor comments:

1)      Why not use absorption at 950 nm instead of 880 nm to estimate BC absorption?

The observed distinctions between these two wavelengths are relatively small. It is also noteworthy that within aethalometers, the wavelength at 880 nm is most frequently employed for the determination of BC, hence reflecting the common practice in assessing BC levels.

2)      Line 395-396: The study defines BrC based on the assumption that absorption above the extrapolated BC absorption with AAE = 1 is attributed to BrC. Therefore, it is no surprise that BrC absorption is correlated with AAE. This statement is circular.

Indeed, this is true. The notion that this relation is expected is now added to the line 402.